# Evolutionary Rim Attention for Linear-Time Sequence Modeling

## Abstract

Transformer self-attention is powerful, but its quadratic dependence on sequence length limits efficiency at long context. Biological nervous systems, by contrast, appear to rely on sparse, local, and hierarchical processing rather than all-to-all pairwise comparison. Motivated by this contrast, we introduce **EvoRimNet**, a sequence model with two parallel pathways: (1) *Inhibitory Rim Attention*, a local competitive operator implemented with multi-scale causal depthwise convolutions initialized with Mexican-hat-like profiles, and (2) *Content-Addressable Thalamic Relay*, a hierarchical sparse memory that supports content-based write and read operations. On WikiText-2 word-level language modeling, EvoRimNet achieves **162.6 test perplexity**, versus 167.3 for a Modern Transformer (RoPE + SwiGLU + RMSNorm) and 188.9 for GPT, improving over both baselines at matched parameter counts across three random seeds. At sequence length 4096, our implementation is **14.8× faster** and uses **67× less peak memory** than the GPT baseline in our measurements. Ablations confirm that each biological component — inhibition, thalamic relay, hierarchical routing — contributes independently and measurably to performance. These results demonstrate that local competitive inhibition combined with sparse content-addressable relay can match or exceed transformer-based attention on language modeling benchmarks, while scaling linearly in both time and memory.

## 1 Introduction

The Transformer architecture (Vaswani et al., 2017) is built around a dominant computational primitive: pairwise interaction between sequence positions. At sequence length $N$, standard self-attention forms an $N \times N$ interaction pattern, producing quadratic time and memory cost in the worst case. This has motivated a broad literature on efficient alternatives, including linear attention (Katharopoulos et al., 2020), state-space models (Gu et al., 2022; Gu & Dao, 2023), and long convolutional sequence models (Poli et al., 2023).

The human brain processes continuous sensory streams — millions of neural signals per second — with no quadratic bottleneck. A cortical neuron connects to at most a few thousand neighbors within a radius of 1–3 millimeters. There is no biological mechanism for every neuron to compare its state with every other neuron. The brain must have solved the $O(N^2)$ problem. The question is how.

**An evolutionary motivation.** Early nervous systems faced exactly this problem. Approximately 540 million years ago, the Cambrian explosion produced the first complex eyes (Parker, 2003). Almost overnight in evolutionary terms, organisms went from sensing gradients of light to processing high-resolution visual scenes. This was the first information crisis in the history of life. The challenge was existential: an organism that could not rapidly distinguish predator from prey, edge from background, would not survive. But the available neural hardware was primitive — sheets of neurons with only local connections, no centralized processor, no global bus. The pressure was not abstract; it was survival. How do you find the important signals in a flood of data when each processing element can only talk to its immediate neighbors?

**Local selection through lateral inhibition.** Evolution's answer was lateral inhibition. In the horseshoe crab *Limulus*, Hartline, Wagner, and Ratliff showed that activation in one receptor suppresses nearby

responses, thereby enhancing local contrast (Hartline et al., 1956). Each neuron excites itself and inhibits its neighbors; what survives is not the average, not a random sample, but specifically the boundaries, edges, and contrasts — the *rims* — where information content is highest. Efficient-coding theory further argues that early processing stages should reduce redundancy before transmitting signals onward (Barlow, 1961). This is not a metaphor for attention. It is an alternative to attention. It answers the same question — "what in this input is worth processing further?" — through competitive suppression rather than pairwise comparison.

**From rims to relay.** Local filtering alone is not enough: selected signals must still be routed to downstream circuits that support integration and action. Comparative accounts of early vertebrate brains suggest that diencephalic and tectal relay structures played an important role in coordinating sensory input with behavior (Feinberg & Mallatt, 2013). In modern vertebrate brains, thalamocortical systems provide a natural example of sparse relay between distributed populations (Sherman & Guillery, 2006). This does not imply that biological brains implement our architecture directly; rather, it suggests that local competition plus sparse relay may be a useful computational design principle.

**Our modeling hypothesis.** Motivated by this perspective, we test the hypothesis that selective information routing can be approximated by two parallel mechanisms: a local competitive pathway that emphasizes informative boundaries and a sparse global pathway that supports content-addressable relay. We instantiate these ideas in EvoRimNet through Inhibitory Rim Attention and a Content-Addressable Thalamic Relay.

**Parallel pathways.** One further biological observation shapes our design. The cortex does not run local processing first and then global relay in sequence. It runs both simultaneously. The dorsal ("where/how") and ventral ("what") visual streams operate in parallel (Goodale & Milner, 1992), and task-positive networks coexist with the default mode network in anti-correlated dynamics (Fox et al., 2005). Inhibition is eliminative: it narrows possibilities by suppressing the unlikely. Relay is generative: it enriches representations by retrieving associated context. One subtracts noise; the other adds signal. This motivates our dual-pathway block, where both mechanisms process the same input in parallel rather than in series.

**Contributions.** We make four contributions:

1. We introduce **Inhibitory Rim Attention**, a local sequence operator with complexity $O(N \cdot d \cdot k)$ for fixed kernel size.

2. We introduce a **Content-Addressable Thalamic Relay**, a hierarchical sparse memory with fixed slot budget and sequence complexity linear in $N$ for fixed memory size.

3. We combine these in a **dual-pathway block** that separates local competition from global relay.

4. On WikiText-2, we show that this architecture can outperform the tested GPT-style and Modern Transformer baselines at similar parameter counts, while providing substantially better scaling at long sequence length in our measured implementation.

## 2   Related Work

**Efficient attention.** Linear attention (Katharopoulos et al., 2020) approximates softmax via kernel decomposition. Performer (Choromanski et al., 2021) uses random feature maps. Linformer (Wang et al., 2020) projects keys and values to lower dimension. These approaches preserve the query–key–value formulation; EvoRimNet instead replaces it with local competition plus sparse relay.

**State space models.** S4 (Gu et al., 2022) and Mamba (Gu & Dao, 2023) provide linear-time sequence modeling through structured state dynamics and selective updates. EvoRimNet differs by combining a local competitive operator with an explicit content-addressable global memory.

**Convolution-based models.** Hyena (Poli et al., 2023) uses long implicit convolutions for efficient sequence modeling. RWKV (Peng et al., 2023) combines recurrent and attention-inspired mechanisms. EvoRimNet uses short-range causal convolutions for local processing and separates global routing into a distinct sparse memory pathway.

**Memory-augmented networks.** Perceiver (Jaegle et al., 2021) and Set Transformer (Lee et al., 2019) use latent bottlenecks or learned queries to mediate interaction. Our relay differs in emphasizing content-based *write* and *read* operations with hierarchical sparse addressing.

**Biologically inspired architectures.** Prior work has drawn inspiration from neuroscience through predictive coding (Rao & Ballard, 1999), plasticity-based learning rules (Miconi et al., 2018), and spiking systems. EvoRimNet focuses on a different combination: lateral inhibition as local competition plus sparse relay as global routing.

## 3 Method: The EvoRimNet Architecture

EvoRimNet is a stack of identical **RimBlocks**, each containing two parallel pathways—Inhibitory Rim Attention and Thalamic Relay—followed by a feed-forward network. The overall structure mirrors the Transformer for direct comparison: tokens $\rightarrow$ embedding $\rightarrow L$ blocks $\rightarrow$ output projection.

### 3.1 Inhibitory Rim Attention

The local pathway operates on an input tensor $\mathbf{X} \in \mathbb{R}^{B \times N \times d}$.

**Step 1: Multi-scale rim detection.** We first transpose to $\mathbb{R}^{B \times d \times N}$ and apply three parallel causal depthwise 1D convolutions with radii $r \in \{3, 7, 11\}$. Each convolution is initialized with a Mexican-hat-like profile:

$$w[c, 1, i] = \delta_{i,c_0} - s \cdot \exp\left(-\frac{(i - c_0)^2}{2\sigma^2}\right), \tag{1}$$

where $c_0$ is the kernel center, $\delta$ is the Kronecker delta, and $s, \sigma$ depend on the scale. Specifically, $\sigma = 0.8 + 0.3 \cdot r/r_{\max}$ and $s = 0.2 + 0.1 \cdot r/r_{\max}$, where $r_{\max} = 11$. All weights remain learnable during training.

The multi-scale responses are averaged and rectified:

$$\mathbf{H}_{\text{rim}} = \text{ReLU}\left(\frac{1}{3} \sum_{r \in \{3,7,11\}} \text{DWConv}_r(\mathbf{X}^\top)\right). \tag{2}$$

**Step 2: Channel mixing.** We then apply a pointwise channel-mixing MLP $(d \rightarrow 2d \rightarrow d)$ with GELU.

**Step 3: Gated residual.** The output is injected through a gated residual:

$$\mathbf{Y}_{\text{loc}} = \text{LN}(\mathbf{X} + \sigma(\mathbf{g}) \odot \mathbf{H}_{\text{mix}}), \tag{3}$$

where $\mathbf{g}$ is a learned per-channel gate.

**Complexity.** For fixed kernel sizes, the local pathway has complexity $O(N \cdot d \cdot k)$.

### 3.2 Content-Addressable Thalamic Relay

The global pathway uses a hierarchical sparse memory and operates in three stages: **Write**, optional **Bind**, and **Read**.

**Memory structure.** Memory is partitioned into $G$ groups of $S$ slots each, for a total of $K = G \cdot S$ slots. Each group has a learned key $\mathbf{g}_j \in \mathbb{R}^d$ and each slot has a learned key $\mathbf{s}_{j,i} \in \mathbb{R}^d$. In the default configuration, $G = 8$, $S = 16$, and therefore $K = 128$.

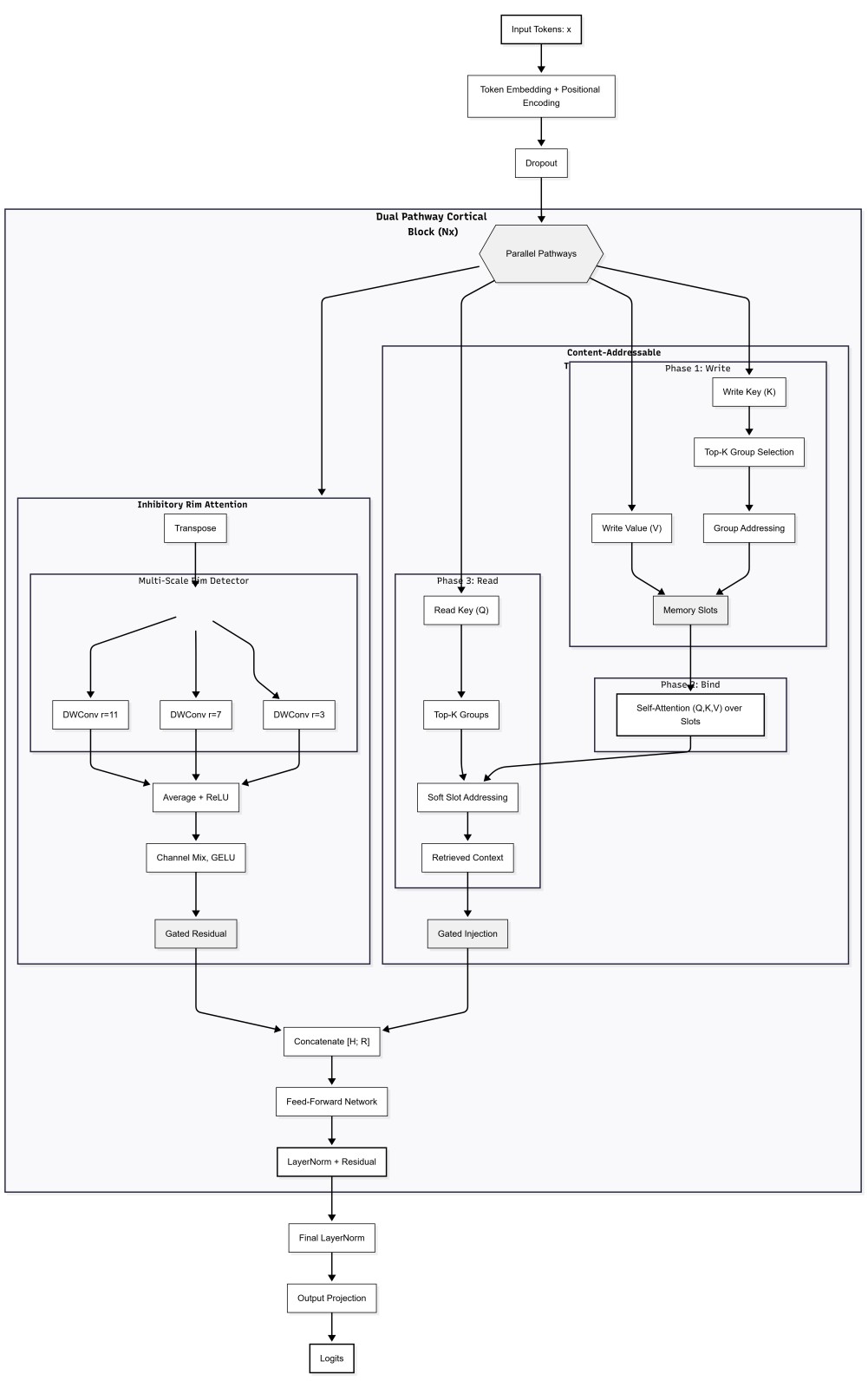

Figure 1: The EvoRimNet Dual-Pathway Block. The architecture processes input tokens through two parallel streams: an Inhibitory Rim Attention pathway for local feature extraction (left) and a Content-Addressable Thalamic Relay for global sparse memory retrieval (right).

**Phase 1: Write.** For each token representation $\mathbf{x}_n$, we compute a write key $\mathbf{k}_n^w$ and a write value $\mathbf{v}_n^w$. Group selection is sparse:

$$\alpha_{n,j} = \text{softmax}_{\text{top-}k}\left(\frac{\mathbf{k}_n^w \cdot \mathbf{g}_j}{\sqrt{d}}\right). \tag{4}$$

Within each selected group, slot weights are computed by a second softmax over slot keys. The resulting memory contents are accumulated with `scatter_add`-style aggregation:

$$\mathbf{m}_{j,i} = \text{LN}\left(\sum_n \alpha_{n,j}\, \beta_{n,j,i}\, \mathbf{v}_n^w\right). \tag{5}$$

**Phase 2: Bind (optional).** An optional slot-level self-attention step can be applied over memory slots between write and read. Our ablations indicate that this phase is not necessary for the best WikiText-2 result.

**Phase 3: Read.** For reading, each token computes a read key and retrieves a weighted combination of group and slot contents:

$$\hat{\mathbf{x}}_n = W_o \sum_{j \in \text{top-}k} \gamma_{n,j} \sum_i \delta_{n,j,i}\, \mathbf{m}_{j,i}, \tag{6}$$

where $\gamma_{n,j}$ and $\delta_{n,j,i}$ are read weights computed analogously to the write weights.

**Gated injection.** The relay output is injected through a learned gate so that the model can adaptively determine how much global information to use at each layer.

**Complexity.** For fixed $G$, $S$, and top-$k$, the relay pathway is linear in sequence length $N$.

### 3.3 Dual-Pathway RimBlock

The two pathways operate in parallel on the same input:

$$\mathbf{H}_{\text{loc}} = \text{InhibitoryAttn}(\mathbf{X}), \tag{7}$$
$$\mathbf{H}_{\text{glob}} = \text{ThalamicRelay}(\mathbf{X}), \tag{8}$$
$$\mathbf{X}' = \mathbf{X} + W_c[\mathbf{H}_{\text{loc}}; \mathbf{H}_{\text{glob}}], \tag{9}$$
$$\mathbf{Y} = \mathbf{X}' + \text{FFN}(\text{LN}(\mathbf{X}')), \tag{10}$$

where $[\cdot; \cdot]$ denotes concatenation and $W_c : \mathbb{R}^{2d} \to \mathbb{R}^d$ is a learned linear projection.

Blocks without the relay pathway use only the local branch. In our default setting, the relay is enabled every other block.

### 3.4 Full Model

The full model consists of token embeddings, learned positional embeddings, dropout ($p = 0.1$), a stack of $L$ RimBlocks, a final LayerNorm, and a linear output projection whose weights are tied to the token embedding matrix. All linear layers and embeddings are initialized from $\mathcal{N}(0, 0.02)$. The thalamic relay gate bias is initialized to $-3$ (so $\sigma(-3) \approx 0.05$), ensuring the relay starts nearly silent and gradually learns when to contribute — mirroring the gradual maturation of thalamocortical connections during development (Kostović & Judas, 2010).

## 4 Experiments

### 4.1 Datasets

- **Shakespeare** (character-level): 1.1M characters, vocabulary size 65, 90/5/5 split.

Table 1: Training configurations per dataset.

| Dataset | seq_len | batch | epochs | batches/ep | lr | $d_{\text{model}}$ |
|---|---|---|---|---|---|---|
| Shakespeare | 256 | 64 | 40 | 80 | $3 \times 10^{-4}$ | 128 |
| WikiText-2 | 256 | 32 | 40 | 100 | $2 \times 10^{-4}$ | 128 |
| enwik8 | 512 | 32 | 20 | 200 | $3 \times 10^{-4}$ | 128 |

- **WikiText-2** (word-level): 2.1M training tokens, vocabulary size 33K (words with count $\geq 3$), official train/validation/test splits.

- **enwik8** (byte-level): 100M bytes, vocabulary size 256, standard 90M/5M/5M split.

### 4.2 Baselines

- **GPT**: hand-written causal Transformer with pre-norm, weight tying, and GPT-2-style initialization.

- **ModernTransformer**: RoPE + SwiGLU + RMSNorm baseline with causal masking.

Both baselines use explicit causal masks rather than `nn.TransformerEncoder` wrappers.

### 4.3 EvoRimNet Variants

- **EvoRimNet (local)**: inhibitory pathway only, without thalamic relay.

- **EvoRimNet (flat 128)**: flat content-addressable relay with 128 slots.

- **EvoRimNet (1 round)**: hierarchical sparse relay, 8 groups × 16 slots, one write–read round, with bind.

- **EvoRimNet (no bind)**: same as above but without the slot self-attention bind phase.

- **EvoRimNet (2 rounds)**: two iterative write–read rounds.

### 4.4 Training Protocol

All models use AdamW (Loshchilov & Hutter, 2019) with $\beta_1 = 0.9$, $\beta_2 = 0.95$, weight decay 0.1, cosine learning-rate schedule with 1000-step linear warmup, gradient clipping at 0.5, and mixed-precision training. Parameter counts are matched approximately within 5% using width search over candidate $d_{\text{model}}$ values. Each experiment runs with three seeds: 42, 137, and 2024. Final perplexity is computed by non-overlapping sequential evaluation on the validation and test sets, using the best-validation checkpoint.

## 5 Results

### 5.1 Language Modeling Perplexity

Table 2 presents the main results across all three datasets.

**WikiText-2 (word-level).** The best EvoRimNet variants achieve **162.6 test perplexity**, improving on both the ModernTransformer baseline (167.3) and the GPT baseline (188.9). The "no bind" variant matches the two-round variant at lower computational cost, suggesting that hierarchical write/read addressing alone is sufficient for this task. Using Welch two-sample t-tests on the three-seed test perplexities, EvoRimNet (no bind; 161.9, 164.4, 161.6) significantly improves over ModernTransformer (166.9, 167.6, 167.5; $t = -5.14$, $p = 0.028$) and GPT (187.8, 189.1, 189.8; $p = 5.09 \times 10^{-5}$). The difference between EvoRimNet (no bind) and EvoRimNet (2 rounds) is not significant ($p = 0.923$), indicating that the second round mainly stabilizes performance rather than improving the mean test perplexity. Although the sample size is limited to three seeds per model, the effect size relative to ModernTransformer is large.

Table 2: Test perplexity (mean ± std over 3 seeds). Best in **bold**, second-best underlined. Parameter counts are per-dataset (width-matched).

| Model | Shakespeare | WikiText-2 | enwik8 |
|---|---|---|---|
| GPT | $6.1 \pm 0.0$ | $188.9 \pm 0.8$ | $6.4 \pm 0.1$ |
| ModernTransformer | $\mathbf{4.7 \pm 0.0}$ | $167.3 \pm 0.3$ | $\mathbf{3.5 \pm 0.0}$ |
| EvoRimNet (local) | $6.5 \pm 0.0$ | $209.7 \pm 1.2$ | $6.4 \pm 0.0$ |
| EvoRimNet (flat 128) | $6.0 \pm 0.0$ | $179.0 \pm 0.9$ | $5.9 \pm 0.0$ |
| EvoRimNet (1 round) | $5.6 \pm 0.1$ | $166.3 \pm 3.0$ | $5.8 \pm 0.0$ |
| EvoRimNet (no bind) | $5.5 \pm 0.3$ | $\mathbf{162.6 \pm 1.3}$ | $5.8 \pm 0.1$ |
| EvoRimNet (2 rounds) | $5.8 \pm 0.1$ | $\mathbf{162.6 \pm 0.4}$ | $5.8 \pm 0.0$ |

Parameter counts are reported as Shakespeare / WikiText-2 / enwik8: GPT (1.2888M / 5.8321M / 1.3603M), Modern-Transformer (1.2272M / 5.3094M / 1.4350M), EvoRimNet (local) (0.8575M / 5.1089M / 0.9147M), EvoRimNet (flat 128) (1.2212M / 5.4726M / 1.2785M), EvoRimNet (1 round) (1.2891M / 5.5405M / 1.3463M), EvoRimNet (no bind) (1.2891M / 5.5405M / 1.3463M), and EvoRimNet (2 rounds) (1.3882M / 5.6396M / 1.4454M).

Table 3: WikiText-2 ablation (test perplexity, mean ± std, 3 seeds). Each row adds one component relative to the row above.

| Configuration | Params | Test PPL |
|---|---|---|
| EvoRimNet (local only) | 5.1M | $209.7 \pm 1.2$ |
|   + flat thalamus (128 slots) | 5.5M | $179.0 \pm 0.9$ |
|   + hierarchical sparse routing | 5.5M | $166.3 \pm 3.0$ |
|   + remove bind phase | 5.5M | $\mathbf{162.6 \pm 1.3}$ |
|   + 2 write-read rounds | 5.6M | $162.6 \pm 0.4$ |
| GPT (reference) | 5.8M | $188.9 \pm 0.8$ |
| ModernTransformer (reference) | 5.3M | $167.3 \pm 0.3$ |

**Shakespeare (character-level).** EvoRimNet (5.5–5.6) improves over GPT (6.1) but remains behind the ModernTransformer (4.7).

**enwik8 (byte-level).** EvoRimNet (5.8) improves over GPT (6.4) but trails the ModernTransformer (3.5). One possible explanation is that the ModernTransformer benefits from RoPE, whereas EvoRimNet currently uses learned positional embeddings.

## 5.2 Ablation Study

Table 3 isolates the contribution of each component on WikiText-2.

The ablation reveals a consistent progression:

- **Inhibition alone** gives a plausible local model but lacks strong global context.

- **Adding flat thalamic memory** substantially improves performance.

- **Adding hierarchical sparse routing** improves performance further.

- **Removing the bind phase** slightly improves performance on WikiText-2.

- **Using 2 rounds** does not improve the mean perplexity but appears to reduce variance.

Table 4: Forward pass latency (ms, $B{=}8$) and peak memory (MB) vs. sequence length. $d_{\mathrm{model}}{=}128$, 4 layers. Measured on NVIDIA A100.

| | Latency (ms) | | Memory (MB) | |
|---|---|---|---|---|
| $N$ | EvoRimNet | GPT | EvoRimNet | GPT |
| 64 | 5.6 | 1.9 | — | — |
| 256 | 5.6 | 1.8 | 26.1 | 41.5 |
| 512 | 5.6 | 3.8 | 33.1 | 102.3 |
| 1024 | 5.7 | 15.3 | 47.3 | 330.5 |
| 2048 | 10.8 | 90.0 | 75.6 | 2250.8 |
| 4096 | 24.5 | 363.3 | 132.2 | 8867.4 |

Table 5: Synthetic task accuracy (%, $d_{\mathrm{model}}{=}64$, 4 layers).

| Model | Pattern Rec. (local) | Majority Vote (global, $N{=}256$) | Retrieval ($N{=}256$) |
|---|---|---|---|
| GPT | 84.7 | 65.9 | 100.0 |
| EvoRimNet (local) | 93.8 | 77.2 | — |
| EvoRimNet (full) | **97.5** | **90.6** | **100.0** |
| No inhibition | 59.4 | 56.2 | — |

### 5.3 Scaling: Speed and Memory

EvoRimNet's latency grows much more slowly than the GPT baseline as sequence length increases. At $N{=}4096$, the measured implementation is **14.8× faster** and uses **67× less peak memory** than GPT. The crossover point is approximately $N{\approx}512$, beyond which EvoRimNet is faster in these measurements.

### 5.4 Synthetic Task Validation

To isolate the contribution of each mechanism, we evaluate on three controlled tasks (Table 5).

**Pattern recognition.** On local pattern recognition, EvoRimNet shows a strong inductive bias for structured transitions.

**Majority vote.** On a global counting task exceeding the local receptive field, the full model improves substantially over both GPT and the local-only variant, suggesting that the relay contributes useful long-range information.

**Long-range retrieval.** On long-range retrieval, EvoRimNet matches GPT at 100%, indicating that the relay can recover specific token-linked information across long distances in this controlled setting.

## 6 Discussion

**Interpretation of the biological analogy.** The central claim of this work is not that biological brains literally implement EvoRimNet. Rather, we test whether a biologically inspired combination of local competitive inhibition and sparse content-addressable relay can serve as a useful alternative to quadratic attention in sequence modeling. The WikiText-2 results suggest that this may be a viable design principle, at least at small scale.

**The "no bind" finding.** Our ablation reveals that removing the slot self-attention bind phase — which corresponds to the CA3 recurrent pattern-completion stage in hippocampal circuitry — slightly improves performance on WikiText-2 (162.6 vs. 166.3). This is not merely a negative result; it may be informative

about the division of labor in biological memory systems. The bind phase models CA3 recurrent connections, which are most strongly associated with episodic memory consolidation — a process that operates over timescales of seconds to hours (Nakazawa et al., 2002). Next-token prediction, by contrast, is closer to working memory: maintaining and retrieving recently encountered items over short spans. The cognitive neuroscience literature distinguishes these systems explicitly (Baddeley, 2003), and our results are consistent with that distinction. The write/read pathway — corresponding to the entorhinal $\rightarrow$ dentate gyrus $\rightarrow$ CA1 route — appears sufficient for the working-memory demands of autoregressive prediction. The CA3 recurrent loop, which evolved for deeper consolidation, adds unnecessary complexity in this regime.

This suggests a testable prediction: on tasks that require multi-step associative reasoning or long-horizon memory integration, the bind phase should become beneficial. We leave this investigation to future work.

**The enwik8 gap and positional coding.** EvoRimNet's weaker performance on enwik8 (5.8 vs. 3.5) is the clearest limitation of the current architecture. The enwik8 corpus contains highly structured XML and HTML markup where precise positional relationships between tags, attributes, and content determine the correct prediction. The ModernTransformer's RoPE provides explicit relative position information that captures these relationships; EvoRimNet's learned absolute positional embeddings do not. This gap is biologically informative. The mammalian brain does not represent temporal order through static positional labels. Instead, the hippocampal formation uses theta phase precession — a code in which a neuron's firing position within an oscillatory cycle carries ordinal information (O'Keefe & Recce, 1993). Incorporating an analogous phase-based or relative positional mechanism into EvoRimNet would maintain the biological grounding while potentially closing the enwik8 gap. This is a natural and well-motivated direction for future work.

**Speed–accuracy trade-off.** At short sequence lengths, EvoRimNet is slower than GPT because the relay introduces constant overhead. Its advantage emerges only at longer contexts. For applications requiring long sequence processing, this scaling behavior may be beneficial.

**Limitations.** This study has several limitations. First, the relay implementation relies on `scatter_add`, which may create unfavorable constants. Second, experiments are performed at relatively small scale ($d_{\mathrm{model}}$=128, roughly 1–6M parameters). Third, the current model lacks a stronger relative-position mechanism. Fourth, the current experiments use simple whitespace tokenization; subword tokenization (BPE) and its interaction with the inhibitory mechanism remain unexplored. Fifth, broader comparisons against additional long-context baselines (Mamba, RWKV, Hyena) on shared benchmarks remain for future work.

**Future work.** Natural extensions include larger-scale training, alternative positional encodings, kernel-level optimization of relay operations, and evaluation on larger corpora such as WikiText-103 or The Pile.

**Broader impact.** This work explores whether biological circuit principles can inform practical machine learning architectures. The linear scaling properties of EvoRimNet may be particularly relevant for domains that require long-context processing under computational constraints — including genomic sequence analysis, clinical time-series monitoring, and document-level understanding. More broadly, we hope this work encourages deeper exchange between computational neuroscience and deep learning: the brain has had 500 million years to optimize information processing under energy and wiring constraints, and its solutions may contain engineering principles that remain underexploited.

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
