# OpenReview forum: "Evolutionary Rim Attention for Linear-Time Sequence Modeling"
_TMLR — Under review for TMLR_

### Review · Reviewer_n1Pw · 2026-04-23

**Summary Of Contributions:**

Evolutionary rim attention proposes an O(N) (N is the number of tokens) sequence modeling architecture - tested on language modeling tasks. The primary objective of this work is to eliminate the O(N^2) computational cost associated with causal softmax attention in modern transformer-based architectures. The authors take a biologically inspired approach of local inhibition and content-addressable relay for global sequence dependencies. To the best of my knowledge, a potential novelty of the proposed architecture is the parallel pathways of both mechanisms. Established O(N) sequence modeling architectures tend to chain such mechanisms (by adding local convolutions before passing the then locally mixed key, value pairs into a global content-addressable key-value memory. Coming back to this work, the architecture is evaluated on three small-scale language modeling tasks, reporting promising results that are somewhat attributable to the architectural components through an ablation study.

Before outlining specific strengths and weaknesses, I must express serious concerns regarding the quality of the submitted manuscript. The document lacks the formal rigor and structured layout expected of a submission to a journal such as TMLR. Notably, the write-up frequently relies on a fragmented, bulleted style rather than providing a coherent, scholarly narrative. Furthermore, the mathematical formalism is insufficient, as the equations describing the architecture are incomplete. The manuscript also lacks a necessary appendix to detail fundamental experimental settings and implementation specifics. Finally, the neuroscientific motivation feels anecdotal rather than analytical, appearing more as a promotional framing than a rigorous theoretical foundation.

That being said, the remainder of my review will be relatively brief. Given the concerns detailed above, it is difficult to grasp the technical nuances of the method necessary to evaluate its overall quality. Should the authors revise their work and provide a higher-quality manuscript, I would be happy to refine my analysis of the paper.

Key-strengths:
- The key architectural novelty is the parallel local and global sequence mixing pathways. Such operations tend to be chained in current $O(N)$ sequence modeling architectures. A particular example: In the DeltaNet (Yang et al., 2024), the keys and values are first mixed by a local 1D convolution and then fed into a global memory state. The authors should consider an ablation that investigates the benefits of the parallelism. On top, an experiment validating the short- vs. long-context behavior of the proposed architecture against a transformer might be nice since recent work, e.g., MesaNet (von Oswald et al., 2025), identifies that many related architectures lose against a softmax transformer in long-context perplexity while being competitive in overall perplexity due to short-context gains.
- The architecture is evaluated on real (while small-scale) language datasets.

Weaknesses:
- The primary weakness remains my concerns regarding overall quality, which I will not restate.
- The related work section appears to be incomplete: From my areas of expertise, I can tell that the literature on, for example, linear attention and its variants is not properly represented by merely citing Katharopoulos et al. (2020).

**Audience:**

No

**Audience Explanation:**

For the reasons outlined above, this question cannot currently be answered in the affirmative.

While this assessment holds for the present manuscript, the core concept of parallel sequence-mixing pathways is intriguing. If more clearly outlined and rigorously validated, this architectural approach could indeed be of significant interest to the TMLR audience.

**Broader Impact Concerns:**

The work at hand resembles general research to advance the field of machine learning. There are no specific impact concerns.

**Claims And Evidence:**

No

**Claims Explanation:**

Due to the quality concerns outlined above, it is difficult to arrive at a well-founded answer to this question. Both the equations describing the architecture and the experimental details lack the rigor and specificity required to judge the quality of the contribution and the results. Furthermore, some of the claims regarding neuroscience do not appear to be grounded in empirical evidence. I am happy to revise my answer if a more rigorous manuscript is provided.

**Requested Changes:**

Please refer to the summary for details on my concerns regarding quality. If the authors address my concerns regarding quality and ideally provide a revised manuscript, I am happy to revisit the paper and provide a refined review of the paper since the overall idea is interesting.

---

### Review · Reviewer_yz6V · 2026-04-23

**Summary Of Contributions:**

In this submission, the authors propose a bio-inspired neural network architecture, called EvoRimNet, which achieves competitive performance on text modeling. In particular, the proposed model is a sequential model with two parallel pathways: Inhibitory Rim Attention and Content-Addressable Thalamic Relay. In small-scale experiments, replacing the multi-head attention layer of Transformer with these two modules can reduce memory costs and improve perplexity.

Strengths:

1. The topic is important. It would be exciting if a strong competitor to the modern Transformer emerged.

Weaknesses:

1. The writing of this submission is questionable, making the motivation of the model design unclear and unconvincing.

1.1 The authors claimed that the model design is inspired by neuroscience. However, without sufficient background and preliminaries, readers (like me) fail to find the similarities between the proposed modules and our brain/neuron architectures. Therefore, the section on related work requires more references and explanations. In the method section, a preliminaries subsection is necessary to explain and illustrate key concepts such as "Rim" and "Thalamic Relay".

1.2 The authors just show the architecture of the proposed model, without any explanations or analysis. From the names of the proposed modules (Inhibitory Rim Attention and Content-Addressable Thalamic Relay), I fail to see the motivations for their design or how they connect to neuroscience.

2. The superiority of the proposed model is not verified because of the lack of large-scale experiments. On WikeText-2, many models can achieve competitive performance compared to the Transformer, especially when the model size is small. However, when scaling to large datasets and model sizes, these models often become inferior to Transformers. At least, the authors should build their model as large as NanoGPT and compare it with NanoGPT on some well-known text-generation benchmarks. Purely considering perplexity is insufficient to demonstrate the usefulness of a sequence model at the current stage.

**Audience:**

Yes

**Audience Explanation:**

See the strength shown in the summary session above.

**Claims And Evidence:**

No

**Claims Explanation:**

See the weaknesses in the summary session above.

**Requested Changes:**

1. The authors should polish their submission substantially, adding more background knowledge, illustrations, and explanations of their model design.

2. Experiments having a scale comparable to NanoGPT's setting are necessary.